# Legume integration in smallholder farming systems for food security and resilience to climate change

Meseretu Melese[1]*, Bereket Getachew[1], Endalkachew Woldemeskel[2], Ashenafi Hailu Gunnabo[1]

1 College of Natural and Computational Sciences, Arba Minch University, Arba Minch, Ethiopia, 2 World Agroforestry (ICRAF), ILRI-Addis Ababa Campus, Addis Ababa, Ethiopia

* meseretu2023@gmail.com; meseretu.melese@amu.edu.et

## Abstract

A random sample of 847 households (HHs) in southern Ethiopia was used to assess the impact of HH characteristics, land and livestock ownership, seasonal variations, and agricultural challenges on cereals and legumes, crucial to community livelihoods. A structured survey was integrated into the ODK data collection tool, validated, and used with trained agricultural agents, and analyzed using an R statistical package. Farm productivity was significantly influenced by gender, since male headed HHs produced 25.5% more yield than female headed HHs. Livestock ownership, land cultivation, farm productivity and product utilizations were also varied among the HHs. Furthermore, agricultural practices like labor utilization, fertilizer and composite application, and use of improved seed varieties significantly affected the farm productivity. Besides, maize was identified as the principal and top-priority crop, while common bean (legume) was the second-priority crop for local people. This shows that legumes were important in the region's agricultural systems, but some farmers experienced reduced productivity due to poor agronomic practices. Consequently, a substantial proportion of farmers (>50%) faced food shortages from February to June, representing the peak of the dry season and the beginning of the rainy season (April to May). Integrating early-maturing legumes in the farming system would help to escape the food shortage periods. Many farmers disclosed that the cost of chemical fertilizers' was unaffordable, indicating a need for options like use of rhizobia inoculants and showed interest to use on their farms for improved productivities.

## Introduction

Smallholder farmers in Sub-Saharan Africa have begun to encounter significant alterations in their environmental conditions, particularly those related to climate changes. The indicators of climate change, including elevated temperatures, extreme drought

**Data availability statement:** All relevant data are within the paper and its Supporting Information files.

**Funding:** The research was funded by the Fund for Innovation and Development (FID) and focused on the adoption of Rhizobium Inoculant Technology (ARIT) at Arab Minch University, Ethiopia.

**Competing interests:** The authors have declared that no competing interests exist.

conditions, the proliferation of pests and diseases, as well as erratic rainfall patterns [1]. The consequences of climate variability resulting a shortage of food, water deficit, outbreaks of infectious disease of both human and animal diseases, and losses of biodiversity which result in a loss in endemic and adaptive food sources [2]. The region's overreliance on rain-fed agriculture, climatic changes such as irregular and unreliable rainfall patterns, and rapidly growing populations [3] pose significant concerns to food supply [4]. To cope with the impacts of climate change, there is a rising interest in the significant restructuring of agricultural systems throughout Africa that enhances farmers' resilience to climate change [5,6].

Agriculture is an important sector in Ethiopia's economy, with a significant contribution to the GDP (32.6%), and export share (77%), and 72.7% of the working population is dependent on it [7,8]. Small-scale farmers, who own less than 2 hectares of land [7] make up 80% of the population in the country and play a significant role in agriculture. These farmers utilize subsistence farming methods and produce more than 95% of the country's crops [8]. The main crops produced by those farmers include cereals, legumes (pulses), oilseeds, vegetables, root crops, fruits, and cash crops like coffee and khat, with suboptimal yields [6].

To reduce vulnerability of smallholder farmers, agricultural intensification is needed to increase the productivity of fragmented plots, maintaining ecological function and increase resilience to shocks [9]. The cultivation of legumes is pivotal in the pursuit of sustainability owing to their capability to assimilate atmospheric $N_2$ through symbiotic relationships with rhizobia; they furnish organic resources and can mitigate other limitations by enhancing soil fertility [10]. The relation among legumes, sustainable intensification, and the improvement of farmers' livelihoods is extensively documented. Moreover, legumes provide organic inputs that positively influence the chemical, physical, and biological properties of soil, thereby enhancing agricultural productivity [9]. Grain legumes constitute a significant category of staple food crops in Ethiopia, ranking second to cereals, and account for 10% and 87% of the overall annual grain output within the nation, respectively [11].

A comprehensive analysis reveals that legumes constitute 12.90% of the total national agricultural land dedicated to farmlands [9]. More than twelve distinct leguminous species are cultivated within the country [14]. Among these, four predominant legume species include faba bean (*Vicia faba* L.) (30%), common bean (*Phaseolus vulgaris* L.*) (19%) chickpea (*Cicer arietinum* L.) (12%) and field peas (*Pisum sativum* L.) (14%) [8]. Smallholder farmers have derived significant advantages from legumes as sources of food, specifically excellent sources of proteins for poor farmers [12] and animal fodder [13], as well as tools for soil nutrient enhancement. Both herbaceous and tree legumes possess the capacity to rejuvenate soil fertility in degraded landscapes and mitigate the processes of land degradation [14].

Despite the country's potential for growing the legumes many Ethiopian soils lack essential nutrients such as nitrogen, phosphorous and potassium which are crop yield limiting factors [15]. Among these soil nutrients, nitrogen (N) is an essential macronutrient for plants, and its availability highly influences the plant productivity [16], needing frequent replenishment. Consequently, prompt resolutions to the issues

pertaining to soil fertility are imperative; chemical fertilizers have historically been advantageous over extended periods; however, their overutilization has resulted in deleterious environmental repercussions [17]. Besides the effect of N-fertilizer on the environment, smallholder farmers often do not afford the prices, have poor access to the fertilizers in their vicinity and consequently produce crops below their potential [18].

Thus, smallholder farmers choose to add legumes as an alternate source of N-fertilizer because of their $N_2$-fixation capability when in symbiosis with soil bacteria called rhizobia [19]. Thus, farmers can fill in geographical and temporal gaps in cropping systems by using intercropping legumes with cereals and rotating their crops [20]. Notwithstanding all the above potential benefits, the area under legume cultivation is small in most farming systems in SSA (Sub-Saharan Africa) [15], with the most prominent legumes being grain legumes because grain legumes have the benefit of supplying immediate returns to smallholder farmers [14]. They can be integrated with all farming systems, as solo crops in rotation with cereals [21], as intercrops between root crops, banana (*Musa paradisiaca*) or Enset (*Ensete ventricosum*) [19] and still the yields vary seasonally and across locations [22]. Their inclusion in the farming system still embraces traditional practices with poor or no input supply.

Though it has been claimed that legume cultivation and agriculture have increased in sub-Saharan Africa due to expansion, it is not sufficient to feed the population. Adding to this, in remote parts of sub-Saharan Africa inputs and modern technologies are less likely accepted as the technologies slowly reach the areas and farmers have poor access to. So, the farmers are left behind. In Ethiopia, there exists significant potential for legume production; however, the actual output and productivity of legumes remain below this potential. This discrepancy can primarily be attributed to various environmental and climatic influences, inadequate utilization of agricultural inputs, restricted access to improved seed varieties, and suboptimal extension services. Consequently, this research endeavor has been undertaken to analyze the determining factors of the production status of legumes among smallholder farmers at the household level. The findings were initiated to generate data for measuring the performance of HHs, farming and management practices, vulnerability to climate change, risk aversion strategies, and enhancing production systems. Furthermore, it evaluates pertinent issues regarding strategies that warrant governmental attention and intervention, as well as promising methodologies in the relevant contexts. Additionally, we sought to evaluate farmers' willingness to adopt novel technologies such as biofertilizers to facilitate future intervention strategies.

## Methodology

### Description of the study area

The study was done at the South Ethiopian regional government which is situated between approximately 6° 51′ 10″ N, 37° 45′ 16″ E. The area encompasses four of the traditionally defined Ethiopian agro-ecologies such as *Kolla*, *Woina-dega*, *Dega, and Wurch* [12]. However, the region's main distinguishing feature is a rich and humid midland, home to Ethiopia's densest rural population. Most districts are densely populated, including the Rift Valley dry plains, such as the Gamo, Konso, and Derashe districts; in contrast, settlements in the South Omo area follow Omo basin and cover a wide geographical area, with sparsely distributed agro-pastoral patches of villages [23].

Agriculture accounts for about 92% of the population's income and home consumption [21]. Except for Konso, and pastoral areas in Omo valley, the enset based farming system is dominant in the highlands of the region. Teff (*Eragrostis teff*), maize (*Zea mays*), barley (*Hordeum vulgare*), wheat (*Triticum aestivum*), faba bean (*Vicia faba*), field pea (*Pisum sativum*), and common bean (*Phaseolus vulgaris*) were grown for both commercial and domestic purposes in lowland and midland altitudes of the region. In the study area, the principal cash crops include banana (*Musa paradisiaca*), coffee (*Coffee arabica*), mango (*Mangifera indica*), sesame (*Sesamum indicum*), cotton (*Gossypium hirsutum*), khat (*Catha edulis)* and apple (*Malus domestica*), tomato (*Solanum lycopersicum*), vegetables and spices. Khat has been expanding and replacing coffee- and enset-based agroforestry systems in the highlands of most of the study area. Among the livestock,

goats, sheep, donkeys, mule, etc. were raised to provide food and income for HHs and used to ploughing land and their manure as natural fertilizer.

Farmland holdings in the region are small, thus farmers use numerous cropping methods, including intercropping and crop rotation, to maximize output per unit area and lower the risk of crop failure due to climate change. Intercropping has become a widespread practice in the research area [26], and farmers make additional revenue while alleviating the food scarcity period. Cereals like maize and sorghum are often intercropped with common beans, faba beans, and field pea [24]. About 57 to 100% of the maize production area is intercropped in early and late belg (small rains) seasons with beans [25]. Field peas and faba beans are commonly grown in rotation with barley, wheat, and potatoes in the highlands. For the current study, we purposely selected potential legume producing districts from Konso, Gamo, Gofa, and South Omo Zones based on the information obtained from regional agriculture office.

### Household (HH) data

The HH survey was conducted during 2022–2023. Information was collected on legume and other crop production and production constraints at the HH level from zones (the administrative structure next to the region) and seventeen districts within the zones. From each zones, two to three districts were purposely chosen, taking into account the area size and population number of the respective districts. Furthermore, the selection process incorporated variations in agro-ecology and the potential for legume production, achieved through consultations with local administrative officials and agricultural expertise from the zones. From each district, three kebeles (the smallest administrative structure) were selected and added to 51 kebeles in two farming seasons and three agro-ecologies (highland, midland and lowland) (S1 Fig).

### Sampling techniques and sample size determination

The study population or farms in the research region were chosen using stratified random sampling. The zones were considered as strata. After determining the required sample size, the proportions used to select the samples from the study area. For determining the sample size for the strata, we adapted the following formula from [26]:

$$n = \left[ \sum \left( N_i^2 * p_i * (1 - p_i) / w_i \right) \right] / \left[ \left( N^2 * e^2 / Z^2\_\{\alpha/2\} \right) + \sum \left( N_i * p * (1 - p) \right) \right]$$

Where n is the sample size needed, N is the total population size, L is the total number of strata (Zone), Z alpha over two is the value of the upper tail probability of the standard normal, distribution Z = 1.96 for α = 0.05 significance level, Pi is the subpopulation proportion of farmers who have knowledge about legume-rhizobium inoculant technology for stratum i and i = 1, 2,…, L, e is the level of precision/marginal error/, Ni is the size of stratum (Zone) i and wi is the estimated proportion of Ni to N Considering N1 = N. Gamo ≤ 35518 N2 = N. Gofa ≤ 35436 N3 = N_S. Omo ≤ 18745 N4 = N. Konso ≤ 26519 N5 = N. Basketo ≤ 1634 N6 = N. Amharo ≤ 6056 e ≤ 0.035 marginal error. It varies from 1% to 5%, p = 0.50, since no research has been done on this issue and there was no information about p. The sample size n is the equation for n = 779. District agricultural experts selected kebeles for a study based on potential and agro-ecology.

### Data collection

Structured questionnaires were designed to collect data on socio-demographic and economic, agricultural, and service-related factors. It also included HH crop production priority (considering legumes, cereals, root crops, vegetables, fruits, and spices), owned livestock, production challenges, and their land holding sizes. The questionnaires were developed in English and then translated into Amharic. Agricultural expertise from districts and agricultural extension workers from each kebele were trained and used for data collection. We used Open Data Kit (https://opendatakit.org), a digital data collection tool and uploaded it to a secured data repository (Kobo Toolbox, https://kobotoolbox.org). In areas where the data

collectors failed to use the tool or had no smartphone, we employed paper-based data collection and later incorporated. In this case, an extra 10% contingency was added to the estimated household number, adding the total number of households to 859. To ensure data quality, incomplete and wrongly filled forms were discarded and the remaining was combined with ODK data. Interviews included questions about all legume crop species produced by the household, followed by more in-depth questions about the target legume, the production constraints, the seasonal effect on the production and intensification method, and their perception of rhizobia inoculant technology.

## Data analysis

The collected data were downloaded from the Kobo Toolbox repository, while data collected on paper base were filtered and manually curated before combining with the data downloaded from the tool. In total, 847 household responses were subjected to descriptive and inferential statistical analysis using R software version 4.3.2. Farm sizes were characterized as small (<1 ha) and large (≥1 ha) by slightly modifying the FAO category [27], while livestock owned by the farmers was expressed in tropical livestock unit (TLU) and categorized as small and large TLU according to Tache et al. [28]. Correlations between demographic variables and farmers' productivity were analyzed using the *agricolae* package in R software.

Similarly, multiple linear regressions were used to evaluate the effect of different farm factors on the farmers' productivity. We employed *the ggplot2* package to visualize the results of the analysis.

## Results

### Demography of the households

To study the factors influencing farm productivity and livelihoods of the farmers, 847 HHs were interviewed, and their responses were analyzed. Accordingly, the ages of the respondents (HHs) ranged from 21 to 83 years of age. The average family size of the HHs was 6.81, representing on average a large family size and a preponderance of rapid population growth in the country. HHs from pastoralist areas had the maximum number of the family size (20), with large farm size holding capacity. The HH size in adult equivalent scale (AES) was estimated to be 4.74. The educational qualifications of the participants were delineated as follows: individuals categorized as illiterate constitute 44.46%, those who have completed primary education account for 39.1%, respondents with secondary education represented 13.22%, and those who have attained post-secondary or university degrees comprised 3.31% (Table 1). Given this, the educational level of the head of the HHs had no effect on agricultural output in the area, since their productivity (in tons) did not differ probably due to lower proportion (3.31%) of the HHs heads with Post-secondary education and above. Relating the demography of the HHs to total

**Table 1. Household characteristics.**

| Variables | Category | No. | Percent | Productivity (in tons) |
|---|---|---|---|---|
| Sex of HH head | Male | 794 | 93.74 | 3.13±0.70 |
| | Female | 53 | 6.26 | 1.86±0.41 |
| Age of the HH | Male | 21-83 | Na | 1.86±0.41 |
| | Female | 30-55 | Na | 3.13±0.70 |
| HH head Educational | Illiterate | 377 | 44.51 | 2.32±0.41 |
| | Primary | 331 | 39.79 | 2.50±0.43 |
| | Secondary | 112 | 13.22 | 2.82±0.51 |
| | Post-secondary | 21 | 2.48 | 2.58±0.87 |
| | University | 7 | 0.83 | 2.23±1.38 |
| HH Size (AES) | Mean of AES | 4.74±0.61 | Na | Na |

Where: HH = households; AES = Household size in adult equivalent scale, Productivity = Mean farm productivity per year, Na = not estimated.

farm productivity (aggregate farm product per year divided by land farmed in that specific year) was not surprisingly differing among the HH educational status. As the level of the HH head education increased, farm productivity did not improve. On the other hand, male-headed HHs had significantly higher mean farm productivity per year per hectare, probably since male-headed HHs had access to farm inputs, more formal interaction with agricultural extension service providers, and become better in adopting varieties & implementing agronomic management (Table 1). The study explores the relationship between family size and crop production in a region, revealing that larger families face challenges in food insecurity due to limited resources sharing, and addressing this requires enhancing cropping intensity and diversification.

In terms of land possessions, some respondents had no arable land and relied on the cottage industry for a living, while the largest land possession reported was 8 ha (Table 2). The farmers annually cultivated between 0.1 and 6 hectares of the arable land mentioned, resulting in significant diversity in both agricultural practices and crop yields. Production records indicated that a maximum of 16 tons was harvested per year, contrasting with the minimal yield of 0.1 tons during the same period, highlighting the wide range of output across farms.

Majority of their output was used for domestic use (in most cases up to 90%) and savings during tough times (common for farmers with surplus farm produce) that have been strongly varied from 10–90% among the HHs. Wealthier farmers or households sold enough to generate income and have modest living and sent their children to school.

Livestock possession was predominantly concentrated in lowland regions, with the highest of TLU (38.52) documented in the pastoralist territories of the South Omo zone. A limited number of individuals residing in the highland areas of the region exhibited a complete absence of livestock ownership, resulting in a pronounced variability in livestock possession at the individual household level. Consequently, the data demonstrated considerable variability, accompanied by exceptionally high standard deviations (Table 2).

## Farm challenges cross the study area

Most people in the study area depended on agriculture for their livelihood, and agricultural goods were their main source of income. Population increases shrunk the farm sizes and made the people farm more fragmented land, affecting their entire productivity. The growing population and its effects on development and agriculture have been the subject of contentious debate locally and globally. In the current study area, there was a conversion of natural forests, sacred forest, and uncultivable areas into farmlands (personal observations during survey of the study sites), indicating the challenges of land shortage and the strategy of the communities to cope with challenges.

## Land fragmentation

The research sites displayed a noteworthy degree of land fragmentation, despite the continual expansion of agricultural territories and the establishment of new settlements in remote regions. Land fragmentation intensity has been defined at the district and zone levels within the research area (Fig 1. tif).

**Table 2. Farm characteristics owned by the households.**

| Variables | Mean | sd | Min | Max |
|---|---|---|---|---|
| Arable land (ha) | 1.71 | 1.21 | 0 | 8 |
| Land cropped | 1.33 | 0.95 | 0.1 | 6 |
| Product in tons/farm/year | 2.89 | 2.94 | 0.1 | 16 |
| Farm product sold (%) | 18.59 | 15.52 | 25 | 75 |
| Farm product consumed (%) | 68.07 | 23.4 | 10 | 90 |
| Farm product saved (%) | 12.02 | 12.78 | 10 | 90 |
| Tropical Livestock Unit (TLU) | 5.2 | 4.22 | 0 | 38.52 |

At the zone level, small-scale farmers in Basketo, South Omo, and at lower altitudes of Gamo zones exhibited larger farmland (>2ha) holdings, while considerable number of farmers experienced farmland fragmentation (<2.0 ha) in Konso, Gamo, and Gofa highlands (Fig 1a, S2 Fig). Not surprisingly, at district level, all farmers cultivated <2ha farm sizes, except for North Ari district (Fig 1b). The allocation of land per household (6.8 persons/HH) was found to be insufficient for food production (S2 Fig). Subsequently, younger HH members started migrating to urban areas, while others were looking for non-agricultural practices.

## Tropical livestock unit (TLU)

In the study area, mixed farming and pastoralist practices were common. Most respondents had a variety of livestock species that constituted local breeds and reared using low-input/low-output husbandry techniques. Residents in pastoral regions had the largest number of tropical livestock units (TLU) compared to HHs living in other lowlands that practiced crop-livestock mixed farming. The highest TLU was estimated for farmers residing in South Omo, where the pastoralists live in the region. Despite the effect of drought (climate impact) on the livestock population in the pastoral area, still a substantial number of TLU was recorded from the area (S2 Fig; Table 2). In the lowlands of Gamo, farmers focused on fruits mainly banana and vegetable productions, and had a small livestock population when compared to pastoralist areas but had a large population when compared with the highland and midland dwellers. Nowadays, in Gamo low lands, the farmers are changing forests and grazing lands to banana plantations, which reduced the number of livestock being reared by the HHs (S3 Fig).

Farmers who possess a significant number of TLU primarily utilizing oxen for ploughing tend to yield a greater number of cereals in comparison to legumes. This can be attributed to the fact that utilizing oxen for ploughing allows farmers to

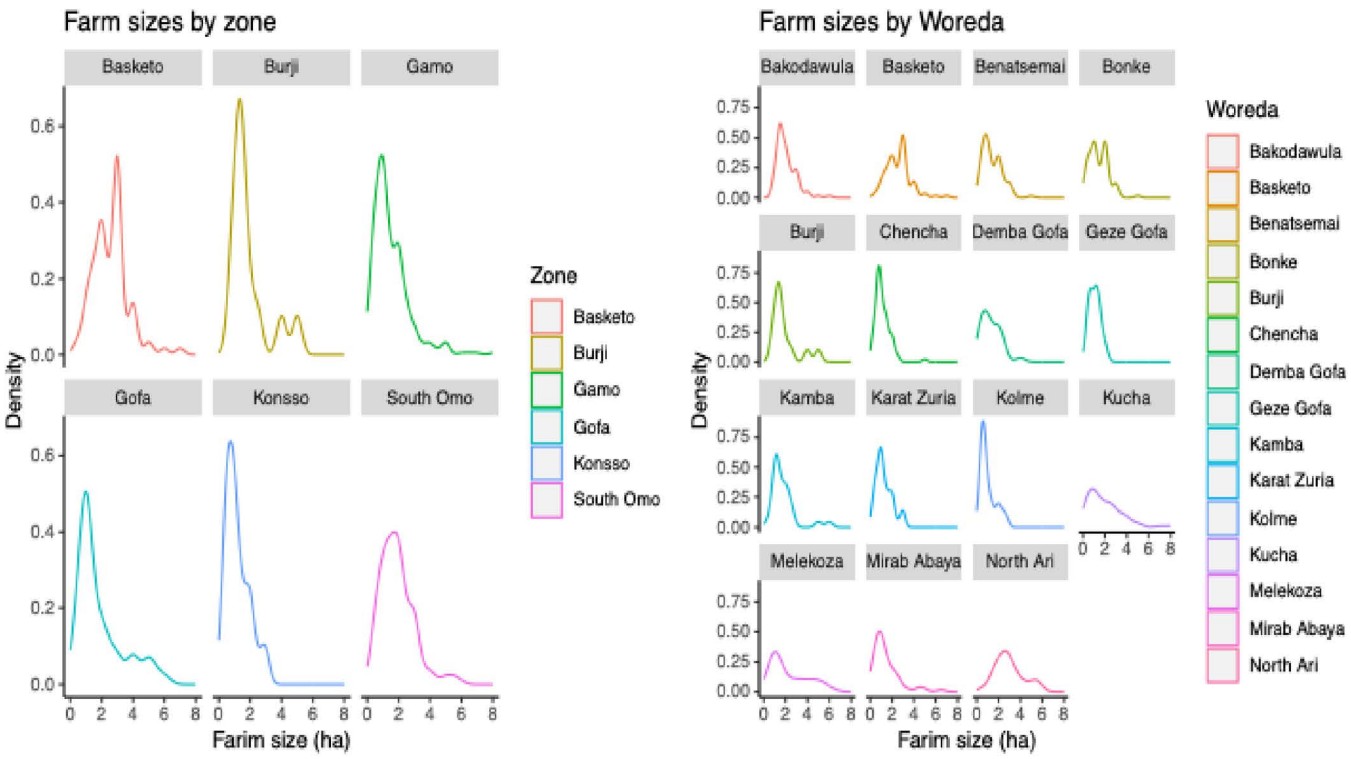

**Fig 1. Farm sizes by zone (a) and by district (b).** shows the average farm size per household in hectares by zone (a) and district (b). The figure illustrates the spatial variation in farm size distribution by comparing mean landholding sizes across districts (subdivisions within zones) and zones (administrative units below the regional level).

effectively cultivate larger areas of arable land, which in turn influences their preference for growing major cereals such as maize, teff, wheat, and barley (S4 Fig).

## Agricultural intensification

To feed the growing national and regional populations, agricultural intensification through organic/chemical fertilizers and improved agricultural practices have been recommended. Accordingly, besides chemical fertilizers, organic fertilizers such as animal manure, compost, green manure and biofertilizers have been in use for over decades. In the current study area, manure application was common among HHs living in highlands, whereas those living in lowlands possessed many livestock but rarely used manure or compost to improve their farm fertility (Table 3).

**Table 3. Farm practices affecting mean farm produce per ha per year.**

| Farm characters | No. | % | Mean farm product (ton/ha/year) |
| --- | --- | --- | --- |
| HHs that hired labor | 219 | 25.86 | 2.43[a] |
| HHs that not hired labor | 628 | 74.14 | 1.84[b] |
| HHs that used organic fertilizer | 377 | 44.51 | 1.98[b] |
| HHs that not used organic fertilizer | 470 | 55.49 | 2.13 [a] |
| HHs that used compost as organic fertilizer | 193 | 22.79 | 1.94[a] |
| HHs that not used compost as organic fertilizer | 654 | 77.21 | 2.10 [a] |
| HHs that used manure as organic fertilizer | 240 | 28.34 | 2.05[a] |
| HHs that not used manure as organic fertilizer | 607 | 71.66 | 2.07[a] |
| HHs that fertilized legumes | 139 | 16.41 | 1.91 [a] |
| HHs that not fertilized legumes | 704 | 83.59 | 2.10 [a] |
| HHs that used urea fertilizer | 653 | 77.10 | 2.11 [a] |
| HHs that not used urea fertilizer | 194 | 22.90 | 1.92 [a] |
| HHs that used DAP fertilizer | 218 | 25.75 | 2.58[a] |
| HHs that not used DAP fertilizer | 629 | 74.26 | 1.87[b] |
| HHs that used NPSB fertilizer | 343 | 40.41 | 1.81[b] |
| HHs that not used NPSB fertilizer | 504 | 59.50 | 2.24[a] |
| HHs that believe their farms have poor soil fertility | 55 | 6.49 | 1.69[b] |
| HHs that believe their farms have moderate soil fertility | 665 | 78.51 | 2.12[a] |
| HHs that believe their farms have good soil fertility | 127 | 14.99 | 1.93[ab] |
| HHs that have hilly farms | 119 | 14.05 | 2.17[ab] |
| HHs that have plain farms | 525 | 61.98 | 1.93[b] |
| HHs that have valley farms | 203 | 23.97 | 2.35[a] |
| HHs that reported occurrence of pests on their farm | 385 | 45.45 | 2.14[a] |
| HHs that reported no occurrence of pests on their farm | 462 | 54.55 | 2.00[a] |
| HHs that reported occurrence of weeds on their farm | 729 | 86.07 | 2.14[a] |
| HHs that reported no occurrence of weeds on their farm | 118 | 19.93 | 1.61[b] |
| HHs that reported occurrence of disease on their farm | 665 | 78.51 | 2.01[b] |
| HHs that reported no occurrence of disease on their farm | 182 | 21.49 | 2.27[a] |
| HHs that have knowledge about rhizobium inoculants | 25 | 3.00 | 2.14[a] |
| HHs that have no knowledge about rhizobium inoculants | 822 | 97.00 | 2.06[a] |
| HHs that reported drought in the farm | 652 | 73 | 2.07[a] |
| HHs that not reported drought in the farm | 195 | 27 | 2.06[a] |
| HHs that used rhizobia inoculant | 0 | 0.00 | 0.00 |

Treatments with the same letter were not significantly different.

Many farmers apply urea, diammonium phosphate (DAP), or blended NPSB to enhance their crop yield and sustain soil fertility (refer to Table 3). A diverse spectrum of chemical fertilizer applications (ranging from 0 kg to 100 kg per hectare) was reported within the region, with some exceptional instances surpassing 100 kg/ha (illustrated in Fig 2. tif).

In general, the predominant proportion of farmers utilized 50 kg of fertilizer per hectare, whereas a minority resorted to using 120, 150, 200, or 300 kg of fertilizer per hectare. An inadequate amount of fertilizers (<50 kg/ha) was observed among farmers from the Gamo zone. Notably, an extraordinary application of 200 or 300 kg of fertilizer per hectare was recorded in South Omo. Despite the reported suboptimal agricultural outputs, the majority of households (77.10%) incorporated urea into their farming practices. As shown in Fig 2, those farmers applied exceedingly small (mostly <50 kg/ha) quantity of fertilizer that might not contribute to the yield.

However, as indicated above, application rates remain significantly below the required level. The HHs from the highlands believed that their farms had good soil fertility, but the yield reported was lower than the farmers who believed that their farms had moderate soil fertility but that productivity could be increased by hiring labor.

The use of organic and chemical fertilizers did not result in significantly higher yields compared to farmers who did not adopt these agricultural practices. However, the application of chemical fertilizers generally enhanced productivity, with the exception of treatments involving NPSB, which did not exhibit a positive effect.

Interestingly, 62% of the farmers had plain land and produced 3.61 tons of mean farm product per hectare, which was 1.05 tons higher yield than the yield from valley lands. Some farmers reported using fertilizers against legumes, but their yield was lower than those who did not use fertilizers. The key objective of this study was to know about the use of rhizobia inoculants in the region and only 3% of the farmers heard about it but they never used it. Some HHs reporting no problems with weeds, pests, diseases, and droughts on their farms and they had better farm products. Briefly, challenges encountered on farms more importantly affected the yield of the farmers than the effects of using fertilizers and employing labor.

## Diversity of crop on farm and farmer's priority crop

It has been a long time since intervention was introduced to farms to improve farm productivity in sub-Saharan Africa. One of such interventions was farm intensification that focused on introducing crop diversification on farms. To understand the diversity of crops growing in smallholder farmers' field and priority crops of the households, farmers were asked to list type of crops ever grown on their farm. More than 20 crops were recorded, of which maize, sorghum, wheat, barley and teff were reported as the first five priority crops on the farm (Fig 3. tiff).

It was recognized that the cereal crops were the primary crops grown on the farms across the study area (with some exceptions) while the legumes become the second priority crops. Among the legumes, common bean was the number one crop among the second and third priority crops. It was succeeded by sorghum, teff, and maize. The dominance of the common bean as the second and third priority crops would reflect its longtime adoption in the region.

The degree of legume intensification within the region was notably suboptimal. Only 16.41% of farmers apply chemical fertilizers in conjunction with local legume varieties (refer to Table 3). Conversely, a respectable number of farmers cultivated improved seed varieties without the application of fertilizers on their farms. Among the improved common bean varieties, Nasir, Red Wolayta, Acos, Hawassa Damue, and Sari119 were identified, whereas the maize varieties BH540, BH541, and BH548, along with the sorghum varieties Mirti Zer, Merera, and Gudina, were acknowledged by the farmers. These improved seed varieties were released by agricultural research centers and subsequently introduced to the farming community by agricultural extension agents. 98% of the respondents confirmed that with improved varieties application of fertilizers boosts crop yields.

## Legume production and potential in the region

The primary objective of this research was to ascertain the potential for legume production and its challenges and evaluate the application of rhizobium inoculant (biofertilizer) in enhancing the quality of legume crops in the study site (Fig 4. tiff.

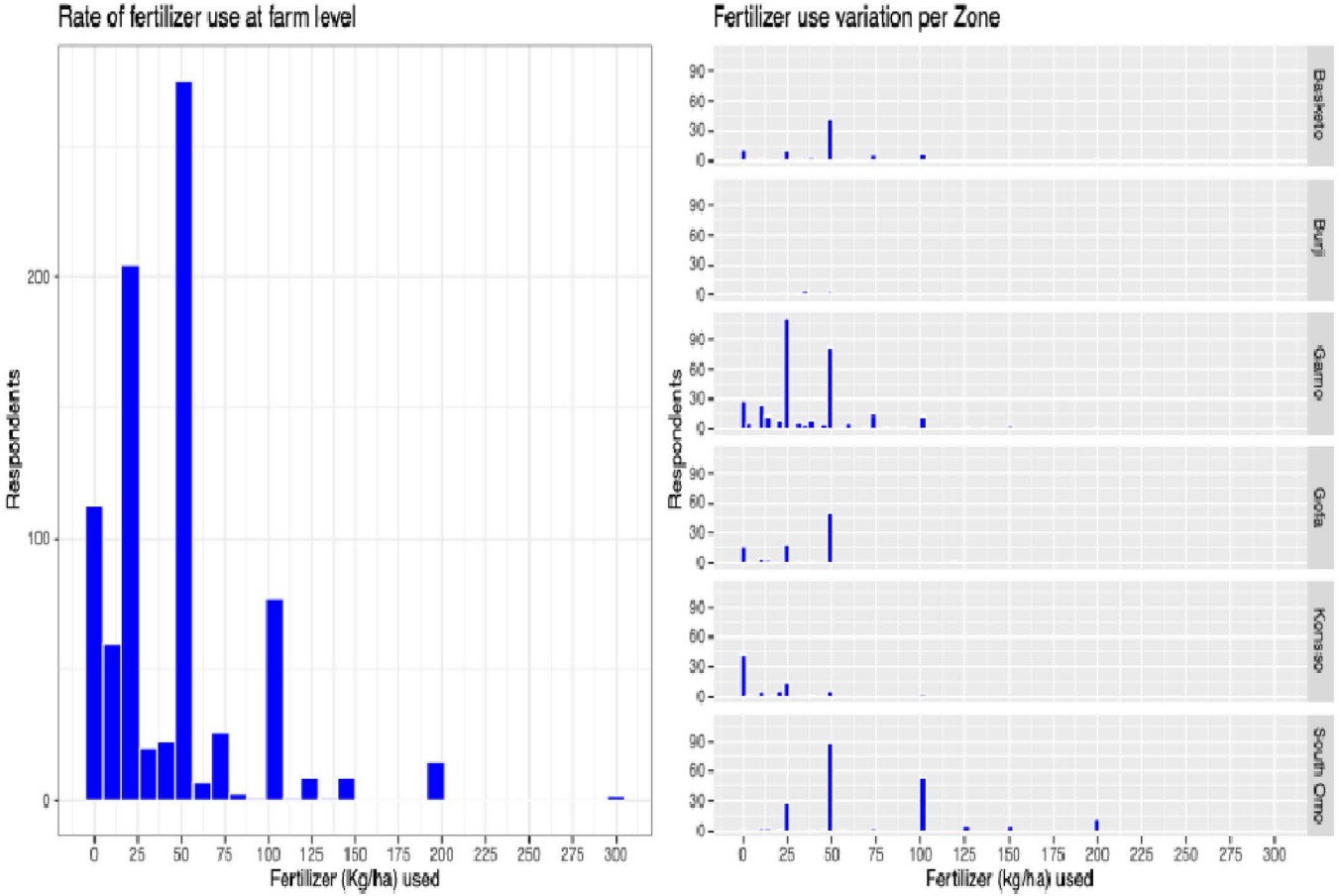

**Fig 2. Average fertilizer application (kg/ha) during a single cropping season across the study area.** And it also shows the mean fertilizer usage per hectare and compares variation in application rates within the zone.

This information was sought to use in the development and execution of legume-rhizobium inoculation technology in the area, aiming to amplify legume yield among indigenous farmers. Thus, the survey investigated legume productivity in the region, finding common bean as the most important legume crop accounting for 45% of the yield. Cowpea follows at 16%, while faba bean and groundnut showed potential for development. Significant legume cultivation was observed across all the three agro-ecologies (*Kola*, *Woinadega* and *Dega*), needing developing a center for their appraisal (Fig 4a).

The majority of those legumes' diversity was recorded in the Gamo zone while a respectable number of them grow in Konso zone. Hence, we further explored the potential of the legumes in Gamo and Konso zones to show the two contrasting cases. In both zones, common bean was the most produced legume (84% in Konso and 41% in Gamo) (Fig 4b, 4c).

Besides common bean, mung bean became the second most important legume crop, which was recently gained the attention of the farmers due to its market value and being one of the Ethiopian Commodity Exchange (ECX) crops. The mung bean has become a commercial crop in Gofa zone and is rapidly expanding and gaining momentum in other parts of the region. However, farmers reported severe damage to the crop by pests and declining market value, demanding for intervention. On the other side, a greater variety of legumes grew in Gamo than in Konso, reflecting variations of legume cultivation between the zones in the region. To that end, only 3% of the farmers know about the rhizobia inoculants but never use it to enhance legume productivity (Fig 4d).

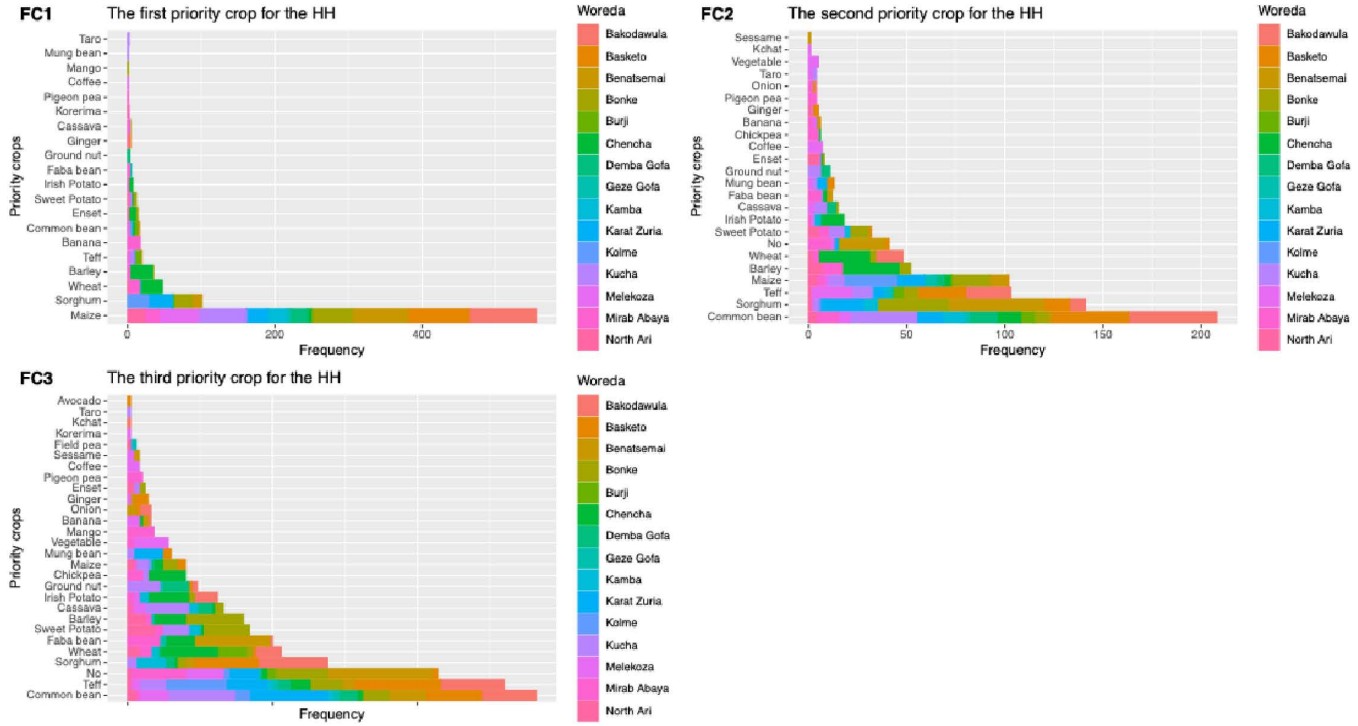

**Fig 3. Crop priority selection at the household level in the research region.** As a reflection of regional preferences for crop production, the figure shows the first, second, and third priority crops chosen by households.

## Farm problems

A number of factors affecting farm productivity have been reported. As such, farmers experienced climate change inducing irregular rainfalls, droughts, incidence of pests and diseases, etc. Majority of the HHs (73%) reported the occurrence of drought on their farm. The drought, exceptionally limited crop production in lowlands of Gamo, South Omo, Konso, and Gofa that faced greater risks of climate fluctuations (droughts and short rainfalls) spilling over several months (Fig 5.tiff). Besides drought, farmland topology induced variations in farm productivity due to runoff. Consequently, farms on sloppy and valley areas had low productivity. Farmers were expanding farms into hilly terrains primarily driven by the limited availability of land, making the problem more intense. Other important factors on the farm include pests and diseases. The abundance of pests varied across zones and destrict, probably linked to agro-climatic conditions of the areas. Nearly 50% of HHs reported pests on farms, while 78% claimed that diseases on their farms (Table 3). The farmers believed that improved seeds would tolerate diseases and pests, and they were requesting for the seeds as well as inputs that they can access at their vicinities (S5 Fig).

## Food shortage months of the year

To explain the food self-sufficiency of the farmers throughout the region, the months of the year that the farmers experience drought and food shortage was assessed. The region experiences two distinct cropping seasons, namely, Meher and Belg, allowing the farmers to produce twice a year. Meher allows the farmers to grow crops from June to September and harvest in October while Belg allows crop to grow from March to May [29].

Accordingly, farmers experienced food shortage over several months from January to July, though the magnitude varied from month to month. From February to June, majority of the farmers (>50%) experienced severe drought and had no

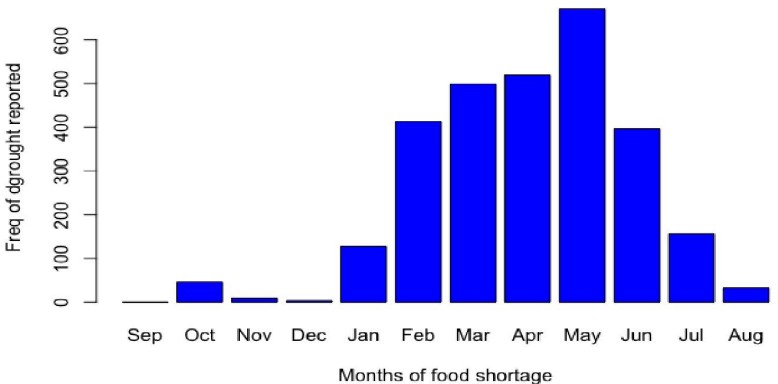

**Fig 4. Legume production potential (4a) and farmer knowledge of rhizobia inoculation in the study area.** The figure compares the most and least diversified legume-producing zones (4b&c) and presents farmers' access to information and knowledge about rhizobia inoculation(4d).

**Fig 5. Food Shortage months in the study area.** shows the study region's seasonal distribution of food shortages. The months with the highest rates of food scarcity (peak hunger months) and those with little to no reported food shortage are shown in the figure. Patterns of food insecurity at the household level throughout the year are reflected in the seasonal trend.

sufficient food to feed the household. The pick of the food shortage was reached in May and thereafter started to sharply drop, since the respondents begun harvesting of vegetables and early maturing crops like legumes such as common beans in June, the end of Belg season. In the region, two distinct rainy seasons such as *Meher* and *Belg* are wet seasons, allowing the farmers to produce twice a year. *Meher* allows the farmers to grow crops from July to September and harvest in October while *Belg* allows crop to grow from March to June. Given those conditions, farmers had sufficient food from July to January, but their farm produce was not sufficient to use in the drier months (Fig 5).

## Discussions

As Sub-Saharan, the country's economy depends on agriculture [30]. Since the country's food production potential did not fulfill domestic demand, it imports a huge amount of food and industrial products that cost billions of dollars. It covers the costs from international donations, remittances, and domestic revenues. To fulfill its food production for self-sufficiency, the country has started massively transforming its agriculture, adopting several new agricultural practices and innovative technologies. Several universities and agricultural research centers have been playing a vital role in adopting and introducing new and innovative agricultural technologies into the country [31]. Rhizobium inoculant technology is one of such agricultural technologies being effectively used in in other countries, but poorly used in Ethiopia [32], particularly in southern Ethiopia. To adopt and use the rhizobium-inoculant technology in smallholder farmers' fields, this study was initiated to generate data for measuring the performance of households, farming and management practices, and vulnerability to climate change and legume production potential. It further wanted to generate first-hand information on the farmer's livelihood and their willingness to adopt the modern technologies like biofertilizers for future interventions. Accordingly, survey data were collected in the regions and described household characteristics, farm variability, agricultural practices, farm problems, climate factors and interest of the farmers towards adopting the rhizobia inoculant technologies.

### Farm variability across the study area

Farm variability refers to the variations in soil composition, crop types, agricultural practices, pest infestations, plant varieties, altitude, soil moisture, soil nutrient levels and harvest outputs from farms within a specific spatial and temporal context. The region's farming systems were distinguished by significant crop-livestock farming systems with high crop diversification and livestock holding. The diverse agro-ecological settings provide for a variety of farming and livelihood systems [34]. Drought, one of the region's climatic factors, has been identified as a significant impediment to people's cropping preferences [36]. Farmers plant a mix of co-staples and annual crops to meet their nutritional needs, and they frequently incorporate income crops [28]. As a result, these farms are mixed subsistence agricultural systems that provide significant self-reliance and long-term existence for these populations [35]. As a result, in the research region, the realization of subsistence systems focused on the extensive cultivation of Enset-Coffee-Khat dominant agroforestry cropping systems throughout the highlands, while maize and sorghum combined with common bean production systems were common in the low and midlands.

Land fragmentation is the growing of crops on disconnected smaller plots [34]. It is more prevalent in the research area, particularly in places with high and medium altitudes, where population growth is high. It is seen as a top concern, and hence its influence has been seen through the lens of farm efficiency and economies of scale [35]. According to CSA [9] and Belay et al. [35], the population boom, customary land inheritance regulations, and farmers' usufruct rights are projected to contribute to increased fragmentation. This suggests that the farms they inherit become incredibly small-scale operations that are incapable of becoming productive even with intensification, and certainly not sufficient for the extension programs that primarily focus on technology diffusion to address the problems of rural poverty. Reduced land size led to considerable reductions in yields of the key crops on which the population relies for food and livelihood security (S7 Fig). Even after a single farming season, respondents revealed that yield reductions fell below their household feed requirements. Because of this, small farm fields did not yield as much, while the fields were farmed on a continual basis

for allowable periods. This caused land degradation, and even fertilizer application was insufficient to improve conditions, and the farmers experienced poor return after fertilizer applications. This coincides with the current findings that fertilization application rates did not bring significant variations farm productivity. Consequently, families of farmers migrate to towns or look for cottage industries.

Farmers in the lowland areas of Gofa, Basketo, and South Omo attempt to increase their cereal crop yield by expanding their farmlands into uncultivated forests, protected areas, and even beyond the boundaries of their villages.

Similar to this, the farmers in Gamo, Konso, Gofa, and Derashe shared uncultivable land with other farmers, even for a single cropping season, in order to utilize all of the fertile ground nearby. This led to a minor increase in cereal production. Farmers in these regions complain that they don't have enough money for emergencies or to buy inputs (S6 Fig). Farmers with small agricultural holdings frequently migrate to neighboring towns for labor jobs as an escape tactic because of their high vulnerability to food and financial instability [45]. Weaving is another choice for the farmers, youths and many HHs in Gamo highlands, an indication of shifting towards cottage industries. In most areas of the highlands, use of animal dung was adapted as a widespread practice of keeping the soil healthy, but as livestock populations declined, smallholder farmers were obliged to use chemical fertilizers. The Enset-Coffee-Khat based farming systems in the highlands had better organic fertilizer application practices than other farmers.

## Agricultural practices and intensification

A considerable proportion of the populace areas in rural regions experience the detrimental effects of malnutrition and persistent food insecurity [33]. Food insecurity was more prevalent in pastoralist areas and dry rift valley lowlands districts than in highlands. This was true for chronic food insufficient areas when prolonged drought occurs. During the period of this survey, the lowlands of Gofa (Demba Gofa and Zala), Gamo (Garda Marta of Kamba district) and Konso (Kolme and Karat Zuria districts) were experiencing severe food shortage due to prolonged droughts that had consequences on their agricultural intensification.

In light of the previously mentioned issue, Ethiopia has made notable advancements in enhancing agricultural output over the recent decades. Nevertheless, despite these positive developments, the levels of yield and overall intensification have continued to be low. In the region, the intensification system has given prominent attention to the distribution and adaptation of chemical fertilizers in ensuring food security but the promotion of improved seeds, which are considered the core of improvement, has less effort from the extension service providers [34]. We observed the extent to which farmers utilized farming technologies and practices commonly recommended by extension services and government authorities.

Regarding chemical fertilizer applications, previously the government had an annual plan to provide fertilizers on a credit base system to the farmers and it was the mandate of agricultural sectors and political leaders because it was considered integral to the sustainable improvement of agriculture [10]. Now, the farmers adopted the chemical fertilizers, and they claim that their farms did not give enough yields without the use of chemical fertilizers. However, the rising costs and in inaccessibility of the chemical fertilizers in their vicinity was severely affecting their farm productivity.

A significant number of smallholder farmers have gained access to improved varieties and technologies through the national agricultural extension system and different technology scaling projects (20). However, it was also observed during the survey and discussion that the farmers were facing seed problems, pests, and disease for which they need improved and resistance varieties.

As the current result shows that farmers were growing diverse crops on the farms, which is a result of intensification, reflecting distribution of improved varieties to the area. Crop diversification is thus one of the farm intensification methods that farmers adopted it in most parts of Ethiopia [35]. The same authors also reported that farmers were adopting the use of improved agricultural practices like increasing on-farm tree planting, soil and water conservation, adjusting planting dates, and crop diversification.

These practices become now an integral part of the farm practices in some parts of the study area, yet others have to adopt them. This is common in the Rift Valley of the current study area where climate change and variability are manifested through frequent droughts and floods, erratic rainfall and fluctuating mean temperatures [36]. So, crop intensification in those areas was less likely practiced.

## Legume production and potential in the region

Legumes are the key component of Ethiopia's smallholder agricultural systems [37]. They are easily adapted farms and used as feed for livestock or as food for humans in a variety of processed forms [38]. These crops are commonly known as "poor man's meat" because they are crucial to the diets of millions of people in developing countries [39]. Grain legumes are among of the most significant commodities in the study region, after cereals making smallholder farmers to produce diverse of them. More than a dozen different legume species have been growing in some parts of the study area. For instance, in Gamo zone, people adopted growing faba beans, field peas, cowpeas, mung beans, common beans, groundnuts, pigeon pea, etc., making the area one of the potential legumes growing sites.

The legumes contribute to smallholder farmers' income as a more valuable crop than cereals, and to diet as a cost-effective source of protein, accounting for around 15% of protein intake and the land coverage increases in all agro-ecology regions [44]. They are common in areas where carbohydrates reach root crops such as cassava, enset, taro, etc. grow and where agroforestry farming practices are common. Southern Ethiopia region is known for agroforestry farming where those legumes become the most important sources of protein for the communities. Studies show that throughout the 1990s and 2010s, there was an increase in the production of legumes, and the export values climbed [42]. The National Statistical Agency reported the area under cultivation and production for the top 10 edible legume crops. In terms of the proportion of the national area production, faba bean (4%), common bean (2.4%), chickpea (1.7%), and field pea (1.69%) are among the major legume crops [11].

Actually, the common bean is one of the Ethiopia's most extensively produced legume crop. It is a valuable source of food, revenue, soil fertility management, medicinal, fodder, and honeybee feed [43]. Their leftovers are used for cattle food and bedding, as well as mulching, fire, and roofing [46]. Its key production areas today include Ethiopia's central, eastern, and southern regions [44]. It thrives in a wide range of environmental circumstances, including well-watered and drought-prone locations between 1000 and 2200 meters above sea level [45]. Smallholder farmers use conventional agronomic approaches to produce common bean crops. It is grown in home gardens and fields throughout its range. It is usually grown for household consumption or sale in local markets, and it has served as a small export crop for over 40 years [43].

Legume cultivation across all agro-climatic zones within the region is quite remarkable. Soybean, cowpea, mung bean, and groundnut are examples of crops that thrive in lowland areas, while other legumes grow in mid-altitude and high-altitude regions. Of particular significance is the widespread growth of common beans across various agro-ecologies, proving to be invaluable to farmers [37]. With a short maturation period of three months, these crops offer a means to escape drought and food starvation times. The potential of short-season grain legumes in addressing community challenges associated with climate variability is substantial, contributing to the development of more robust and productive agricultural systems. Often serving as emergency sustenance for vulnerable HHs, legumes encounter various obstacles in their production [38].

In general, legumes are the most vital component of the farming systems in the region next to cereals (for example, common bean is the second and third most priority crop for the HHs) but no inputs are used for their cultivation. Collectively, their overall productivity is low, needing interventions for improving. The recent development of massive rhizobia inoculant applications against the grain legumes in central parts of Ethiopia has revealed significant improvement of their productivity (www.N2Africa.com). Seeing their potential in the region, escaping environmental risks, growing in different agro-ecologies and recent inoculant application developments that confirmed the improvement of the legume productivity,

we suggest that smallholder farmers need sustainable intensification of the legumes for resilience to climate change and food security in Southern Ethiopia.

## Legumes as an alternative crop for the copping plan

The legume sector in Ethiopia has the potential to be a key accelerator of agricultural development and growth. It can potentially contribute to a valuable role not only in boosting export incomes but also in enhancing the rural economy, improving food/nutrition diversity, enhancing soil fertility, and livestock productivity, reducing global warming and creating more sustainable and climate-resilient food systems due to its ability to fix atmospheric nitrogen through biological nitrogen fixation [39]. Despite legume crops encompassing both edible and fodder crops, and a potential diverse crop portfolio in Ethiopia, the national statistical agency [7] highlights the area under cultivation and production only for the major 10 edible legume crops. So, the national area and total production coverage for major legume crops (faba bean, field pea, chickpea, lentil, grass pea, soya bean, fenugreek, mung bean and lupine) in 2021 were 12.9% and 9.4%, respectively. In terms of the proportion of the national area production, faba bean (4%), common bean (2.4%), chickpea (1.7%) and field peas (1.69%) are among the major legume crops [7]. To increase productivity per plot of land and overcome the ever-increasing price of chemical fertilizer, biofertilizer technology has been gaining attention among agronomists and soil scientists because of its considerable benefits, especially in sustainable agriculture [40].

The country has identified a tremendous potential in biofertilizers to increase the productivity of soils [41]. Even though the country has started to expand the research on biofertilizers, to identify and isolate, more strains of beneficial microbes [40] and a few private companies started to produce and distribute to the farmers, it is limited to the central part of the country. In the study region, the agricultural experts and development agents (DAs) have information about biofertilizers, but the target farmers did not have information about it. This tells us a considerable number of farmers receive or do not at all receive agricultural innovations. Agricultural intensification has been characterized as 'increasing average labor or capital inputs on a smallholding, either cultivated land alone or cultivated and grazing land, to enhance the value of output per hectare' [38], one of such intensification is use of inoculants, which is absent in the current study area.

Fertilizers, improved seeds, agrochemicals, and agricultural gear are among the measures used to intensify land use [39]. Ethiopia has a long history of agricultural intensification through the adoption of technologies that either save labor (the ox-plough), preserve natural resources (various land structures, such as terracing and the use of tree crops to preserve soil integrity), or maximize value per hectare [40]. However, a very rapid population increase offers the demand for intensification, while market access gives the opportunity [41]. Fertilizers are one of the key productivity-enhancing inputs widely promoted by the extension system in Ethiopia to increase yields by addressing the productivity losses caused by declining soil fertility [38].

Fertilizer intensification has been considered a key game changer in Ethiopia's agriculture transformation agenda, and as a result, fertilizer imports have more than doubled over the last two decades [42]. From June 2022 to January 2023, the Ethiopian Ministry of Agriculture (MoA) claimed that more than 15 million quintals of fertilizer were given for a single cropping season, accounting for around 83 percent of the total need. Fertilizer costs, fertilizer prices are still prohibitively expensive for most farmers with poor household incomes because worldwide fertilizer prices are high due to the indirect effects of conflicts, retail fertilizer prices have doubled to 1,400 USD/MT [43]. Farmers and development agents confirm it that when landholding declines, so does per capita legume production. The farmers replied that when they face a shortage of land, they prioritize their main stable cereals and tuber crops (S7 Fig).

Continuous cereal-based mono-cropping practices, soil fertility depletion and erosion problems, limited improvement and farmers' preferred legume varieties, lack of/limited access to quality seeds and no/or minimal uses of inputs in legume production, limited promotion of appropriate legume technologies, wrong farmers' perceptions for legumes input uses are the most important constraints of legume intensification in the country [44]. In addition, problems of institutional setup, limited availability of value-adding processing machines for legume products, policy, and strategy gaps on legume-based

intensification; inappropriate market information and unclear legume market outlets, weak public and private partnerships across each legume commodity are also common bottlenecks.

An alternate approach to the constraints is to apply bio-fertilizer to the soil; this promotes plant growth by rhizobacteria and nitrogen fixers [38]. Bio-fertilizer technique is low-cost and environmentally friendly. There are constraints limiting its application and implementation in the region; these constraints were technological, infrastructural, financial, environmental, human resources, unawareness, and quality.

### Farm problems and food shortage months of the year

Ethiopian soils are experiencing loss of organic matter, biodiversity, fertility by water and wind erosion, and soil acidification. Besides, continued traditional farming of crops mines nutrients from the farm fields without replacement, causing soil nutrient depletion. Many Ethiopian soils become now depleted or acidified and gives poor crop yields. Hence, food production is a persistent and critical issue in the country [42]. Rural poverty and food insecurity pose challenges for millions of individuals across the country. Despite a historical drop, a significant minority of Ethiopian HHs (25%) remain food insecure and vulnerable (27.08%) [43].

In Ethiopia, the interplay between rising temperatures and erratic rainfall patterns—hallmarks of climate change—has profound implications for food security, particularly given the nation's heavy reliance on rain-fed agriculture. Over the past few decades, Ethiopia has seen a noticeable increase in mean annual temperatures. Temperature increases in Ethiopia have accelerated evapotranspiration, resulting in decreased crop yields, increased water stress, and deficiencies in soil moisture. Food security in households has suffered as a result of these climate changes' negative effects on agricultural productivity (44). Rainfall patterns in Ethiopia significantly impact food security, as agriculture, which accounts for over 85% of the labor force, is primarily rain-fed. Changes in rainfall patterns, such as delayed initiation and early cessation, affect crop yields and livestock output, leading to increased food insecurity (45) (S1 Table).

According to FAO [47], the situation has deteriorated as the country has experienced five consecutive failed rainy seasons, the worst desert locust infestation, and the outbreak of conflict in northern Ethiopia, causing many rural households to lose productive assets and their vital resources. Furthermore, drought has claimed the lives of 6.8 million animals in the country's south. These effects have resulted in low crop output, revenue loss, and high rates of malnutrition and food insecurity [48]. Farmers in the study region described drought, diseases, pests, and weeds as the prominent farm factors that contributed to reducing farm productivity (S5 Fig). Climate change causes drought, incidence of pests and diseases and reduces crop productivity [47] that severely affects the livelihoods of the farmers.

### Conclusions

This research advances and provides extensive information on HH demographics, farm variability, agricultural productivity, related problems, and the opportunities and challenges of legume production. Field visits and farmers' interview result analysis identify the productivity of smallholder farmers. Accordingly, the level of farm intensification, household possession of the tropical livestock unit, and the knowledge and accessibility of technologies affect farm productivities in the region. Other food security determinant factors in the area include cultivable land fragmentation at the household level, varied farm topography, and incidences of pests, diseases, and drought in the farms. Notable land fragmentation with frequent breakdowns of pests and diseases reduces sufficient food production. Besides, rain-fed farming systems, characterized by unusual & erratic rainfall and frequent drought, lead to hunger and loss of both crops and livestock. In such conditions, farm intensification was influenced by the shortage of farm water, the accessibility and prices of chemical fertilizers, and the knowledge and availability of biofertilizers. Even though diverse legumes are growing in the area, a negligible (3%) number of HHs heard about the use of biofertilizers, and none of them use the biofertilizers. In conclusion, our findings reflect that creating awareness about the use of legumes and rhizobia inoculants can significantly improve crop production by the smallholder farmers. Along with this, the provision of improved seed varieties that exhibit resistance to

diseases, pests, and drought at affordable prices with implementation strategies is needed to enhance farmers' response to climate shocks. Thus, the production of legumes ought to be integrated into agricultural extension initiatives by facilitating training and supplying biofertilizers within their localities.

## Supporting information

**S1 Fig.  Sampling sites of the study area.**
(TIF)

**S2 Fig.  Household size effects the production.**
(TIF)

**S3 Fig.  Land size fragmentation and tropical livestock unit at zone and district levels in the study area.**
(TIF)

**S4 Fig.  Legume productivity as influenced by tropical livestock unit.**
(TIF)

**S5 Fig.  Factors affecting farm productivity in the study region.**
(TIF)

**S6 Fig.  Farm productivity as influenced by land cropped, arable land and tropical livestock unit .**
(TIF)

**S7 Fig.  Legume and cereal productivities influenced by land cropped.**
(TIF)

**Table S1.  Pearson correlation between demographic and farm factors.**
(TIF)

## Acknowledgments

The authors would like to thank Arba Minch University, for providing facilities and equipment's and ARIT-FID- France for financial support.

## Author contributions

**Conceptualization:** Meseretu Melese Mekuria.

**Formal analysis:** Bereket Getachew Mamo.

**Funding acquisition:** Ashenafi Hailu Gunnabo.

**Investigation:** Meseretu Melese Mekuria.

**Methodology:** Meseretu Melese Mekuria.

**Resources:** Ashenafi Hailu Gunnabo.

**Software:** Ashenafi Hailu Gunnabo.

**Supervision:** Endalkachew Woldemeskel, Ashenafi Hailu Gunnabo.

**Validation:** Bereket Getachew Mamo.

**Writing – original draft:** Meseretu Melese Mekuria.

**Writing – review & editing:** Endalkachew Woldemeskel, Ashenafi Hailu Gunnabo.

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
