## [Decision Letter · Decision Letter 0]

16 Sep 2024

Dear Dr. Mekuria,

Thank you for submitting your manuscript to PLOS ONE. After careful consideration, we feel that it has merit but does not fully meet PLOS ONE’s publication criteria as it currently stands. Therefore, we invite you to submit a revised version of the manuscript that addresses the points raised during the review process.

**ACADEMIC EDITOR:**

We look forward to receiving your revised manuscript.

Kind regards,

Charles Odilichukwu R. Okpala, PhD

Academic Editor

PLOS ONE

Journal Requirements:

3. You indicated that ethical approval was not necessary for your study. We understand that the framework for ethical oversight requirements for studies of this type may differ depending on the setting and we would appreciate some further clarification regarding your research. Could you please provide further details on why your study is exempt from the need for approval and confirmation from your institutional review board or research ethics committee (e.g., in the form of a letter or email correspondence) that ethics review was not necessary for this study? Please include a copy of the correspondence as an ""Other"" file.

4. Thank you for submitting the above manuscript to PLOS ONE. During our internal evaluation of the manuscript, we found significant text overlap between your submission and previous work in the [introduction, conclusion, etc.].

Please revise the manuscript to rephrase the duplicated text, cite your sources, and provide details as to how the current manuscript advances on previous work. Please note that further consideration is dependent on the submission of a manuscript that addresses these concerns about the overlap in text with published work.

[If the overlap is with the authors’ own works: Moreover, upon submission, authors must confirm that the manuscript, or any related manuscript, is not currently under consideration or accepted elsewhere. If related work has been submitted to PLOS ONE or elsewhere, authors must include a copy with the submitted article. Reviewers will be asked to comment on the overlap between related submissions (http://journals.plos.org/plosone/s/submission-guidelines#loc-related-manuscripts).]

We will carefully review your manuscript upon resubmission and further consideration of the manuscript is dependent on the text overlap being addressed in full. Please ensure that your revision is thorough as failure to address the concerns to our satisfaction may result in your submission not being considered further.

5. Please provide additional details regarding participant consent. In the ethics statement in the Methods and online submission information, please ensure that you have specified (1) whether consent was informed and (2) what type you obtained (for instance, written or verbal, and if verbal, how it was documented and witnessed). If your study included minors, state whether you obtained consent from parents or guardians. If the need for consent was waived by the ethics committee, please include this information.

6. Please update your submission to use the PLOS LaTeX template. The template and more information on our requirements for LaTeX submissions can be found at http://journals.plos.org/plosone/s/latex .

7. Thank you for stating the following financial disclosure:

“The research was funded by the Fund for Innovation and Development (FID) and focused on the adoption of Rhizobium Inoculant Technology (ARIT) at Arab Minch University, Ethiopia.”

8. Thank you for stating the following in the Acknowledgments Section of your manuscript:

“The study was supported by Fund for Innovative Development (FID) (ARIT-FID Project No. CET112201) through the ARIT-FID project of Arba Minch University. The authors thank the Arba Minch University for technical and logistical support, FID France for financial support.”

 “The research was funded by the Fund for Innovation and Development (FID) and focused on the adoption of Rhizobium Inoculant Technology (ARIT) at Arab Minch University, Ethiopia.”

9. Please provide a complete Data Availability Statement in the submission form, ensuring you include all necessary access information or a reason for why you are unable to make your data freely accessible. If your research concerns only data provided within your submission, please write "All data are in the manuscript and/or supporting information files" as your Data Availability Statement.

10. Please amend either the title on the online submission form (via Edit Submission) or the title in the manuscript so that they are identical.

11. We note that you have included the phrase “data not shown” in your manuscript. Unfortunately, this does not meet our data sharing requirements. PLOS does not permit references to inaccessible data. We require that authors provide all relevant data within the paper, Supporting Information files, or in an acceptable, public repository. Please add a citation to support this phrase or upload the data that corresponds with these findings to a stable repository (such as Figshare or Dryad) and provide and URLs, DOIs, or accession numbers that may be used to access these data. Or, if the data are not a core part of the research being presented in your study, we ask that you remove the phrase that refers to these data.

12. We note that [Figure 1] in your submission contain [map/satellite] images which may be copyrighted. All PLOS content is published under the Creative Commons Attribution License (CC BY 4.0), which means that the manuscript, images, and Supporting Information files will be freely available online, and any third party is permitted to access, download, copy, distribute, and use these materials in any way, even commercially, with proper attribution. For these reasons, we cannot publish previously copyrighted maps or satellite images created using proprietary data, such as Google software (Google Maps, Street View, and Earth). For more information, see our copyright guidelines: http://journals.plos.org/plosone/s/licenses-and-copyright.

13. We notice that your supplementary [Supplemental Table 1] are included in the manuscript file. Please remove them and upload them with the file type 'Supporting Information'. Please ensure that each Supporting Information file has a legend listed in the manuscript after the references list.

Additional Editor Comments:

Please, kindly attend to the comments made and provide very detailed responses.

Thank you

Reviewers' comments:

Reviewer's Responses to Questions

**Comments to the Author**

1. Is the manuscript technically sound, and do the data support the conclusions?

Reviewer #1: Partly

Reviewer #2: Yes

2. Has the statistical analysis been performed appropriately and rigorously?

Reviewer #1: No

Reviewer #2: Yes

3. Have the authors made all data underlying the findings in their manuscript fully available?

Reviewer #1: Yes

Reviewer #2: Yes

4. Is the manuscript presented in an intelligible fashion and written in standard English?

Reviewer #1: Yes

Reviewer #2: Yes

Reviewer #1: 1.The title is lengthy, clumsy, and very difficult to comprehend. Because there are many independent variables that can be an article alone

2. In terms of cost it is clear that inorganic fertilizers are costly however do you articulate that in terms of productivity and accessibility

3. What is your policy recommendation in general and farm-level recommendation

4. The introduction part is more general and it is all description, importance but you did not supported with strong empirical and conceptual testimonies, even some of them looks like a lecture note

5. What you are going to address is not clear or your objective is not clear

6. In your methodology,I have not seen if there are missing data because you add 10% as contingency, you said nothing how to measure food security status of the households, I have not seen any strong model to see the effect of the mixed farming on household food security status

7. It is better to support the result of demographic data with national standards and the implications

8. Food security will be very difficult to measure using self-sufficiency, what quantity is sufficient, please see food security interns of its pillars.

9, You conclusion lacks clarity, be specific and answer your research questions, it is broad and looks like abstract please revise it

10. It is not also strongly linked with climate change resilience

In general, I appreciate the effort made, however, the paper lacks details of linking crop-livestock system to food security, it should be measurable and have a model to show this. besides I did not see about sustainability issue but boldly written in your title.

Reviewer #2: Line 129 Please correct font size of Gossypium hirsutum as is seen larger as compare to other text

Line 130 make Botanical name of tomato Solanum lycopersicum in italics

Line 479 Please correct the stud with study

**Do you want your identity to be public for this peer review?** For information about this choice, including consent withdrawal, please see our Privacy Policy

Reviewer #1: **Yes: ** Shishay Kahsay Weldearegay

Reviewer #2: **Yes: ** Dr. Narendra Chaudhary, National Research Centre on Seed Spices, INDIA

---

## [Author Response · Author response to Decision Letter 1]

29 Oct 2024

We included the Arba Minch University's ethical review form, which was completed by the research reviewers before accepting the project.(We attached as other 3 pages)

---

## [Decision Letter · Decision Letter 1]

5 May 2025

Dear Dr. Mekuria,

Thank you for submitting your manuscript to PLOS ONE. After careful consideration, we feel that it has merit but does not fully meet PLOS ONE’s publication criteria as it currently stands. Therefore, we invite you to submit a revised version of the manuscript that addresses the points raised during the review process.

**Please attend to the comments in the attached documents in detail.**

We look forward to receiving your revised manuscript.

Kind regards,

Charles Odilichukwu R. Okpala, PhD

Academic Editor

PLOS ONE

**Journal Requirements:**

**Additional Editor Comments:**

Authors, thanks for your patience. Please, you can see reviews are positive, but require additional inputs to improve your work. Kindly attend to the comments in great detail

Reviewers' comments:

Reviewer's Responses to Questions

**Comments to the Author**

Reviewer #1: All comments have been addressed

Reviewer #3: (No Response)

2. Is the manuscript technically sound, and do the data support the conclusions?

Reviewer #1: Yes

Reviewer #3: Partly

3. Has the statistical analysis been performed appropriately and rigorously?

Reviewer #1: Yes

Reviewer #3: N/A

4. Have the authors made all data underlying the findings in their manuscript fully available?

Reviewer #1: Yes

Reviewer #3: Yes

5. Is the manuscript presented in an intelligible fashion and written in standard English?

Reviewer #1: Yes

Reviewer #3: No

**Reviewer #1: ** Still i am not satisfied with the food security measurement applied and the linkage with resilience to climate change. I prefer if it would have been consider the temperature and rainfall in relation to food security. with out considering the temperature and rainfall trends considering other parameter to see the linkage between food security and resilience to climate change is incomplete.

**Reviewer #3: ** The abstract is informative and captures key findings effectively; however, there are minor grammatical inconsistencies, awkward phrasing, and issues with tense consistency that need revision for clarity and professionalism.

It is recommended to clarify abbreviations on first use, refine vague expressions (such as "higher levels"), and maintain consistent past tense throughout. Strengthening the transitions between findings would also improve the abstract’s overall flow and readability.

Please make the recommended changes to the following lines to improve clarity, grammatical accuracy, and consistency:

• Revise the sentence on gender and productivity to specify "higher productivity levels" and clarify resource utilization.

• Adjust the phrasing about agricultural practices to maintain past tense ("some were found to negatively affect productivity").

• "some have been found to affect it negatively" → "some were found to affect it negatively" (tense consistency).

• "farmers have disclosed" → "farmers disclosed" (simpler, more formal past tense preferred).

Conclusion

The content provides valuable insights into household demographics, farm variability, and legume production challenges. However, I suggest merging the two paragraphs into a single coherent paragraph to improve the flow and readability. Additionally, please consider revising and splitting the following long sentence to enhance clarity:

• The sentence starting with "Even if, the region was rich in growing diverse crops..." (lines 631–632) is too lengthy and could be broken into two sentences.

Overall, the findings are relevant, but these improvements will make the presentation much clearer.

Results

Given that higher educational attainment among household heads did not lead to improved farm productivity, what factors might be limiting the effective translation of education into practical agricultural outcomes in this region?

The manuscript mentions that farmers who applied fertilizers did not necessarily achieve higher yields compared to those who did not. However, this statement lacks clarity and supporting data. Please revise this section to clearly explain whether the lower yields were statistically significant and specify whether factors such as crop type, fertilizer type, or application rates were controlled for in the comparison.

In the section discussing land fragmentation, you report both farmland expansion and shrinking farm sizes due to population growth, which appears contradictory. Please clarify this point: are farmers expanding into new lands while their individual farm holdings are simultaneously becoming smaller due to subdivision among heirs? A clearer explanation is needed to avoid confusion.

Methodology

The methodology is clearly described; however, could you clarify how data quality was ensured when combining digital and paper-based data collection methods? Specifically, were there any verification or cross-checking steps taken to minimize inconsistencies between the two approaches?

Reference

Some references are older (e.g., 2014–2017). Consider including more recent studies (2020–2024) to strengthen the background.

**Do you want your identity to be public for this peer review?** For information about this choice, including consent withdrawal, please see our Privacy Policy

Reviewer #1: **Yes: ** Shishay Kahsay Weldearegay PhD)

Reviewer #3: **Yes: ** Dr. Fahmida Sultana

---

## [Author Response · Author response to Decision Letter 2]

24 May 2025

Points raised by academic editors/reviewers

Author response

Reviewer #1: Still, I am not satisfied with the food security measurement applied and the linkage with resilience to climate change. I prefer if it would have considered the temperature and rainfall in relation to food security. without considering the temperature and rainfall trends, considering other parameter to see the linkage between food security and resilience to climate change is incomplete

Dear Reviewer, thank you for your concern, as noted. We integrated the link between rising temperatures and the consequences of rainfall variability on food security, as mentioned in lines 604-614.

Reviewer #3: The abstract is informative and captures key findings effectively; however, there are minor grammatical inconsistencies, awkward phrasing, and issues with tense consistency that need revision for clarity and professionalism

Thank you for your input. We strive to remove grammatical problems online 16-27, 30, 32, 34, and 36 and completely improve the abstract part.

It is recommended to clarify abbreviations on first use, refine vague expressions (such as "higher levels"), and maintain consistent past tense throughout. Strengthening the transitions between findings would also improve the abstract’s overall flow and readability.

Thank you for your input. It was rephrased on lines 18-20

• Adjust the phrasing about agricultural practices to maintain past tense ("some were found to negatively affect productivity").

We appreciate your effort, and we rephrased it from lines 459-488.

Some have been found to affect it negatively" → "some were found to affect it negatively" (tense consistency).

It was rephrased from lines 25

"Farmers have disclosed" → "farmers disclosed" (simpler, more formal past tense preferred).

Re Rephrased on line 33

7. Revise the sentence on gender and productivity to specify "higher productivity levels" and clarify resource utilization.

It was rephrased from lines 15 to 17, 247

8. Conclusions

The content provides valuable insights into household demographics, farm variability, and legume production challenges. However, I suggest merging the two paragraphs into a single coherent paragraph to improve the flow and readability. Additionally, please consider revising and splitting the following long sentence to enhance clarity:

• The sentence starting with "Even if, the region was rich in growing diverse crops." (lines 631–632) is too lengthy and could be broken into two sentences

We truly value your recommendation, which was rephrased in one paragraph.

9. Results

Given that higher educational attainment among household heads did not lead to improved farm productivity, what factors might be limiting the effective translation of education into practical agricultural outcomes in this region?

It was justified from lines 219 to 221

10. The manuscript mentions that farmers who applied fertilizers did not necessarily achieve higher yields compared to those who did not. However, this statement lacks clarity and supporting data. Please revise this section to clearly explain whether the lower yields were statistically significant and specify whether factors such as crop type, fertilizer type, or application rates were controlled for in the comparison

The concept is clarified further, as restated in lines 314 to 323

The supporting data was reanalyzed on table 3,

11. In the section discussing land fragmentation, you report both farmland expansion and shrinking farm sizes due to population growth, which appears contradictory. Please clarify this point: are farmers expanding into new lands while their individual farm holdings are simultaneously becoming smaller due to subdivision among heirs? A clearer explanation is needed to avoid confusion

The justification was added from line 468 to 470

Also, we restated in discussion part from line 509-513

12. Methodology

The methodology is clearly described; however, could you clarify how data quality was ensured when combining digital and paper-based data collection methods? Specifically, were there any verification or cross-checking steps taken to minimize inconsistencies between the two approaches?

We add the justification from line 179 to 182

13. Some references are older (e.g., 2014–2017). Consider including more recent studies (2020–2024) to strengthen the background.

We carefully reviewed your suggestion regarding references, and it was substituted and updated on reference numbers 3, 7, 8, 10, 12, 13, 14, 21, 26, 27, 28, 33, 34, 36, and 39.

---

## [Decision Letter · Decision Letter 2]

20 Jun 2025

Legume integration in smallholder farming systems for food security and resilience to climate change.

PONE-D-24-28412R2

Dear Dr. Mekuria,

We’re pleased to inform you that your manuscript has been judged scientifically suitable for publication and will be formally accepted for publication once it meets all outstanding technical requirements.

Kind regards,

Charles Odilichukwu R. Okpala, PhD

Academic Editor

PLOS ONE

Additional Editor Comments (optional):

Thank you authors for your careful revision of your work. It is acceptable for publication

Reviewers' comments:

Reviewer's Responses to Questions

**Comments to the Author**

Reviewer #3: All comments have been addressed

2. Is the manuscript technically sound, and do the data support the conclusions?

Reviewer #3: Yes

3. Has the statistical analysis been performed appropriately and rigorously?

Reviewer #3: Yes

4. Have the authors made all data underlying the findings in their manuscript fully available?

Reviewer #3: No

5. Is the manuscript presented in an intelligible fashion and written in standard English?

Reviewer #3: No

Reviewer #3: (No Response)

**Do you want your identity to be public for this peer review?** For information about this choice, including consent withdrawal, please see our Privacy Policy

Reviewer #3: **Yes: ** Fahmida Sultana

---

## [Editor Report · Acceptance letter]

PONE-D-24-28412R2

PLOS ONE

Dear Dr. Mekuria,

I'm pleased to inform you that your manuscript has been deemed suitable for publication in PLOS ONE. Congratulations! Your manuscript is now being handed over to our production team.

Kind regards,

on behalf of

Dr. Charles Odilichukwu R. Okpala

Academic Editor

PLOS ONE